# Development of a Procedure for Torsion Measurement Using a Fan-Shaped Distance Meter System

**DOI:** 10.3390/s23208603

**Published:** 2023-10-20

**Authors:** Martina Goering, Thomas Luhmann

**Affiliations:** Institute for Applied Photogrammetry and Geoinformatics (IAPG), Jade University of Applied Sciences, Ofener Str. 16/19, 26121 Oldenburg, Germany; luhmann@jade-hs.de

**Keywords:** fan-shaped distance meter system, camera, photogrammetry, sensor fusion, relative orientation, wind turbine, rotor blade torsion

## Abstract

Maximising the efficiency of wind turbines is crucial for sustainable development of renewable energy. In this context, monitoring and optimising rotor blade performance is becoming increasingly important, especially rotor blade deformation and torsion. We developed an approach for marker-free and contactless measurement of rotor blades during operation. Deformations of rotor blades can be recorded, with focus on torsion measurement. An innovative measuring system, named the fan-shaped distance meter system (FDMS), uses a combination of multiple laser scanners and photogrammetry. The focus of this work is to analyse the suitability of the FDMS for torsion measurement. We designed a torsion simulator to assess the achievable accuracy. Computer simulations and initial laboratory tests have demonstrated precise torsion measurements are possible using this method with an accuracy of 0.3°. Measurements can be carried out during operation of the wind turbine without the need to apply markers or sensors on rotor blades. By precisely recording the deformation and, in particular, torsion of rotor blades, targeted optimisation measures can be obtained in order to maximise performance of wind turbines. This innovative approach to measure the torsion of rotor blades in operation might offer great potential to increase the efficiency and life cycle of wind turbines.

## 1. Introduction

Wind turbines perform an increasingly important role in the generation of electricity from renewable energy sources, contributing to the reduction of greenhouse gas emissions. These turbines are made up of various components, of which rotor blades have a central role. They capture wind energy and convert it into rotational energy that is subsequently converted to electrical energy using a generator.

In addition to an optimal aerodynamic design, rotor blades are able to adapt to varying wind speeds by adjusting their pitch to achieve efficient energy harvest. During operation, rotor blades are subjected to different loads that cause deformations, such as bending and torsion. Bending occurs along the longitudinal axis of the blade and is caused by aerodynamic forces acting on the blade surface. For the outer tip of rotor blades, a deformation in the wind direction (flapwise) of 10% of the blade length is typical [1]. Blades have become significantly longer since 2009 [1] and have changed in shape and material. In addition to bending, torsion is important, as it has a major influence on the performance and service life of wind turbines. Torsion describes the rotation of the blade around the longitudinal axis. The exact magnitude of torsion depends on several factors, including wind speed, the aerodynamic profile of the rotor blade, rotor blade stiffness, and the position along the blade. Due to changes in the aerodynamic profile and loads, torsion generally increases from the rotor blade root towards the tip. In general, torsion at the outer tips of the rotor blades is expected to be in the range of 0 to 10 degrees. To optimise rotor blades, some have a pre-curved design to achieve the optimal shape at full load [2]. By deliberately introducing torsion into the rotor blade system, vibration and unwanted loads are reduced, resulting in increased turbine efficiency and an extended life time.

Both deformations, torsion and bending, are not isolated from each other, but occur coupled. Thus, bending–torsion coupling describes the interaction between bending and torsion [3]. Due to dimensions of today’s wind turbines and high dynamic loads, they represent an extremely demanding object of measurement. In context, comprehensive investigation of deformations of rotor blades are crucial to further improve the performance and reliability of wind turbines and to ensure long-term usability. Deformation measurement sensors installed in blades are complex to use; moreover, they do not last the entire life time, while older blades do not have any sensors installed at all [4,5,6,7,8,9]. Alternatively, optical methods with high-resolution cameras placed inside rotor blades can be used to capture the surface structure within the blade [10,11]. These optical methods provide high-precision measurements and can be used on large rotor blades.

It is also possible to simulate deformations; however, they rely heavily on attached sensors. Therefore, the need for a reliable blade health monitoring system, which is independent of these sensors, is high [12]. A special interest is on linking deformation measurement with wind conditions [13].

Optical 3D surveying provides methods for contactless deformation measurements. It was previously proven that photogrammetry in particular can be used as health monitoring system [14]. Earlier measurements on a real turbine with a rotor blade diameter of 10 m were performed with a stereo camera system in publications of [15,16,17,18,19]. Therefore, the wind turbine was equipped with retroreflective dot markers to record vibrations of the tower and rotor blades simultaneously. However, focus was on the vibration and deformation of the rotor blades without torsion. The deformation behaviour of aerodynamic surfaces of a wing was also performed at discrete points [20] and with a random pattern [21]. An ongoing project on a real turbine using random pattern signalisation should be able to establish a sufficient number of measurements to derive torsion [22,23]. By marking wind turbines, highly precise photogrammetric measurements can be collected. At a new research wind farm, the rotor blades are equipped with a random pattern [24]. However, marking requires long downtimes of turbines for application and removal of the marking, which is not desired by operators. Therefore, marker-free approaches are preferred.

Photogrammetry also offers marker-free surveying capabilities by observing prominent points or lines on the facility even during operation [25]. Marker-free photogrammetry cannot achieve high precision over larger distances, making laser measurement technologies a viable solution. A laser scanner uses a rotating mirror to deflect the laser beam, and 3D coordinates can be calculated in conjunction with the measured distance. Depending on deflection of the mirror around one or two axes, a laser scanner can be used in 1D, 2D or 3D mode. Established geodetic surveying can be used to record tower movements and vibrations [26,27,28,29]. Some companies use two distance meters to measure the distance between the rotor blades and tower, as well as relative deformation of the blades [30]. A combination of laser measurement technology and photogrammetry enables more than 1 degree precise measurements [25]. To derive geometric information from kinematic measurement data, additional sensors can be used to detect movements and perform corrections. In order to be able to draw conclusions on efficiency and material fatigue, associated wind conditions are usually simultaneously recorded using wind lidar sensors.

For detection of vibrations at rotor blades at discrete positions, Ref. [31] presented a method using a laser doppler vibrometer. By determining the position of rotor blades using a camera system and tracking the laser doppler vibrometer with a pan-tilt head, a continuous measurement can be made at the same position. A similar method using a laser scanner in 2D mode and a camera is described in [32,33]. In [34], different approaches for optimisation and ground-based monitoring of turbines are presented.

Measuring torsion of rotor blades is more complex, due to the rotation of blades and the larger measurement volume. In addition, the movement of tower and nacelle overlap with absolute positions and deformation of the blades, so torsion measurements still remain an open research topic.

In our previous work on the topic of non-contact and marker-free measurements of rotor blades (with a focus on torsion) is presented in [35,36,37]. This method uses a distance meter, and the surface of the rotor blade is scanned depending on the angle of incidence and blade shape. This scanned surface is referred to as a “profile” in the further course, even if it is not perpendicular to the longitudinal axis Z (Figure 1, blue line). Torsion is determined based on these data. For this purpose, it is necessary to record at least two profiles. One is measured directly at the hub of the rotor blade to determine the angle of attack, while the other profile is at the tip of the blade. By measuring the relative orientation of both profiles during operation, the torsion φ at the corresponding position can be derived by comparison with the reference. With additional profile data, a detailed resolution of the deformation can be achieved. Four Z + F Imager 5010 [38] laser scanners are used for these measurements as they support both 3D mode and 1D mode. Data on the orientation of laser scanners in 3D mode are recorded, while the profile data on deformation measurements are used in 1D mode as a distance meter. Blade deflection in the wind direction (X-direction) is shown in yellow in Figure 1.

Manufacturers require knowledge on the position of the profile in the direction of longitudinal axis Z to an accuracy of 0.5 m (Figure 1, green markings). It is possible to transform the data into the coordinate system of the rotor blade. At the outer tip, the blade has a blade depth of 1 m. If a torsion φ of 1° around the centre of the blade is assumed, the distance measurement must achieve an accuracy of 8 mm (blade depth of 0.5 m) using the arc formula (Figure 1, side view). These requirements can be met with laser scanners or distance meters.

All distance meters are aligned to the height of the hub and distributed along the blade (Figure 2). The green point measures directly at the hub, and the red point measures at the outer tip. Further points (here in blue and yellow) can be positioned in between. Due to rotation of the rotor blade (rotation angle α), different distances are detected using the distance meters, which depend on the rotation angle α. This enables an angle-based assignment of the measured values. In 1D mode, distances are detected according to the rotation angle of the rotor blades or the time in the measurement system. Data recording at the hub takes longer than at the outer tip because the blade shape and rotation speed are different. Intensity values of the detected laser beam are used to filter the data.

In practical tests, four laser scanners (Imager 5010 from Zoller + Fröhlich) were used on a real wind turbine with a hub height of 100 m and a rotor blade length of 60 m. Measured values on the rotor blade have a standard deviation of the distance measurement of less than ±2 mm (compare with [35]). High precision of the distance measurement enables a torsion determination with an accuracy of less than 1°.

In order to optimise the laser scanner approach, a novel measurement system has been developed called a fan-shaped distance meter system (FDMS). The concept is shown in Figure 3. Four distance meters (blue rectangle) are used for this purpose, and the laser beams (red arrows) form a plane. This system aims to simplify the alignment of the laser beam to rotor blades, improves precision of the orientation, and provides a cost-effective alternative to laser scanners. Details about the FDMS are presented in [37].

Four distance meters from the Z + F Imager 5006 series are used for the FDMS (Figure 3 and Figure 4). Distance measurements are performed using the phase comparison method and have a specified distance measurement accuracy of <1 mm under ideal conditions.

The combination of FDMS with photogrammetry enables the transformation of measured data into a superior coordinate system. In addition, it allows the determination of additional data (e.g., speed of torsion simulator) or the derivation of geometry data of the rotor blade photogrammetrically. To implement these functions, two high-speed cameras pco.dimax CS3 [39] with a base length of 1 m are attached to the frame of the FDMS. Cameras run synchronously with the FDMS, and when an image is captured, a time stamp is stored in the FDMS data. At close range, the laser dot is visible in the camera images.

For calculation of 3D coordinates, it is necessary to determine the relative orientation of the distance meters with respect to each other. Two methods are presented in [31], whereby the method with a moving test field has prevailed due to its simpler handling and evaluation. The procedure is now briefly presented. The first step is to determine the interior and relative orientation of the cameras. For this purpose, photos of a test field are acquired, which represent typical calibration images [40]. Subsequently, image sequences of the test field are taken, showing laser points of the FDMS at different recorded distances. A best-fit straight line is calculated through photogrammetrically determined 3D coordinates of the laser points. The origin and orientation of distance meters along the line are calculated, using measured distances of the FDMS. In this way, six degrees of freedom (X, Y, Z, ω, φ, κ) are determined for each distance meter. With a known relative orientation, recorded distances can be converted into 3D coordinates using a line as specified in Equation (1). For this purpose, the origin of the distance meter (X, Y, Z) is added to the direction vector (ω, φ, κ) multiplied by the measured distance *s*.
(1)$${\vec{l}}_{n}=\left[\begin{array}{c} X\\ Y\\ Z \end{array}\right]+s*\left[\begin{array}{c} \omega \\ \varphi \\ \kappa \end{array}\right]$$

The aim of this study is to present a method to determine the torsion of a rotor blade with the FDMS within an accuracy of 1 degree. Simulations and laboratory tests are described to quantify orientation and torsion measurements. In addition, a comprehensive precision assessment is performed.

Section 2 presents the concept for the torsion measurement and the simulation concept for assessing the achievable precision. Practical implementation of the measurement setup and results are presented in Section 3. Discussion of the results is given in Section 4. The final section summarises the findings and provides an outlook for further research.

## 2. Torsion Measurement

The FDMS is designed to measure objects in motion. As an application, the torsion *φ* of wind turbines during operation is to be detected. In the following, we present the concept and simulation for testing the capability of the FDMS for torsion measurement at wind turbines.

### 2.1. Concept

In order to analyse whether the FDMS could detect torsion at a specific precision level, a torsion simulator was required on which a rotation angle could be set to represent torsion. Similar to a wind turbine, the torsion simulator needed to be recorded in motion for the FDMS. To simplify the evaluation process, the movement could initially be uniform, and rotation around a certain point was not required.

Therefore, a torsion simulator with four planes was developed that had a common axis of rotation (Figure 5, black line). Each plane could be rotated around this axis. During the measurement of a data set, these planes were fixed while the torsion simulator was moved uniformly in vertical direction (green arrow). Each plane was scanned with one distance meter.

Since measured distances lay on laser line $l_{n}$, they had to be rectified to correctly reproduce the torsion simulator (Figure 6). Distances could be corrected using a known velocity ${\vec{v}}_{move}\;$ and the direction vector ${\vec{d}}_{move}$ of the movement. Result are the points ${\vec{p}}_{n}$ on the plane. Thus, it was possible to convert 3D coordinates into a metric and time-independent system.

A best-fit line $\vec{c}$ could be determined using the calculated coordinates ${\vec{p}}_{n}$ (Figure 7). The calculated direction vector of this best-fit line described one direction of the plane. We defined another direction vector parallel to the rotation axis $\vec{r}\;$ of the torsion simulator. Using these two vectors, the normal vector of the plane $\vec{n}$ could be determined using cross product.

If the normal vector was available for each plane, the angle of rotation *φ* between two planes $\vec{n_{i}}$ and $\vec{n_{j}}$ could be calculated using Equation (2).
(2)$$\varphi =\arccos\left(\frac{\;{\vec{n}}_{i}{}^{\circ}{\vec{n}}_{j}}{\left|{\vec{n}}_{i}\right|⬝\left|{\vec{n}}_{j}\right|}\right)$$

To verify the method and measurements, photogrammetric reference data could be acquired under laboratory conditions (maximum recording distance of 10 m). For this purpose, the torsion simulator was provided with targets. These targets could be tracked during movement of the object such that a plane could be determined for each time, from which a normal vector could be derived in each case. Furthermore, tilt sensors were used to verify the method.

### 2.2. Simulation

The calculation procedure for determining the angle of rotation *φ* was analysed using simulated data for the laboratory setup. The simulated recording distance was set to 10 m. For this purpose, defined planes were moved in 3D space along a vector, and corresponding measurements of the FDMS were calculated. Various input parameters were modified with noise, and a Monte Carlo simulation was performed with 10,000 runs. In the simulation, the standard deviation of the respective rotation angles *φ* of the planes was determined.

The parameters of the relative orientation of the FDMS were varying within their standard deviation, with s_0_ a-posteriori of 1 mm for XYZ and 0.03° for rotation angles. This had only a small influence of 0.007° on the result of the rotation angle *φ*. The simulation process indicates that the position of sensors within the FDMS has no significant influence on the result. The uncertainty of the relative orientation with 2σ (position and orientation) resulted a linear increase in the standard deviation to twice the rotation angle *φ* (0.014°).

Since the angle of rotation *φ* was derived from the distance differences, these have a high influence. This could be confirmed in the laboratory setup. At 10 m, a distance noise of 0.5 mm led to an angular noise of 0.06°, with 1 mm distance noise to 0.12°. There again was a linear relationship, and it showed how relevant the distance measurement was for torsion determination. In a simulation with a recording distance of 150 m and a distance noise of 2 mm (compare to Section 3.1), a noise of torsion angle of 0.2° was calculated. This shows that torsion determination with the required accuracy was possible using the FDMS.

## 3. Practical Implementation

This section describes the practical implementation in the laboratory. This is followed by a comparison of the calculated normal angles between the FDMS and the reference system. The following workflow was required for torsion determination in the laboratory setup (Figure 8).

### 3.1. Measurement Setup and Data Acquisition

The torsion simulator consisted of four planes with a common axis of rotation. In order to minimise disturbing factors on the distance measurement and to focus on the determination of the angle of rotation *φ*, the torsion simulator should consist of planes with a uniform white surface. Each plane had a size of 200 mm × 500 mm. The torsion simulator was mounted on a lift truck, which allowed slow movements. Figure 9 and Figure 10 present the torsion simulator.

**Figure 8 sensors-23-08603-f008:**
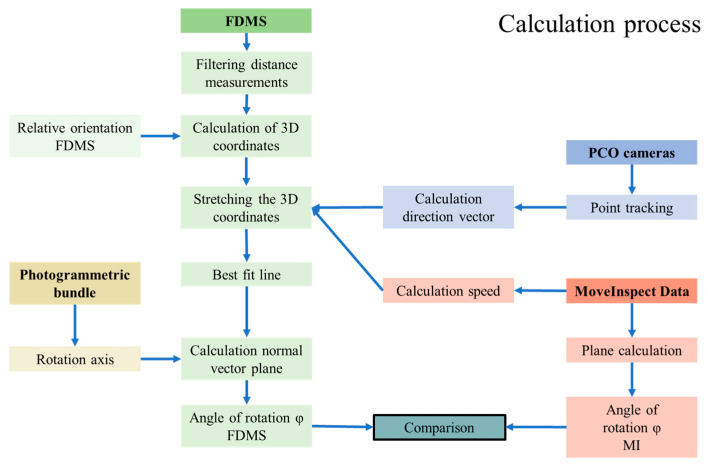
Calculation process.

Photogrammetric targets were attached to the torsion simulator. Retro-reflective points were measured and tracked automatically. From the measured values of individual points, the direction vector of the torsion simulator was determined using a best-fit line; likewise, the speed of the torsion simulator was obtained. An AICON MoveInspect system (MI) was mainly used to determine comparative data of the rotation angles *φ*. For this purpose, eight measuring points per plane were considered, which were located next to the measuring path of the laser (Figure 9, red rectangles). These were used to determine the direction and speed of the torsion simulator. Standard deviation of the object coordinates was less than 0.01 mm after a photogrammetric adjustment, so these data could be used as a reference. Furthermore, the angle of rotation *φ* was calculated by fitting a best-fit plane through the measurement points describing the plane. The standard deviation of the normal vector within a measurement series showed a maximum value of 0.1°. The angle of rotation *φ* between the planes was calculated directly from the respective normal vectors.

The FDMS and PCO cameras were set up with a recording distance of 10 m (Figure 11). The FDMS was set to measure with a minimum frequency of 32 kHz. PCO cameras recorded images with 100 fps. For synchronisation, a signal was sent to FDMS during image recording. The images of the PCO cameras were processed using the software of AICON MoveInspect Pilot 7.11.13. In addition, an optical measurement system, AICON MoveInspect HF 4 high frequency, was used to record 3D coordinates of retroreflective targets at 100 fps synchronously to the other systems. The MoveInspect system with three cameras aligned on a common base achieves an accuracy of 0.1 mm in a measuring volume of 1 m^3^ and, therefore, presented a measurement system of higher accuracy for comparison. The MoveInspect was set up with a distance of 3 m. It could be used to independently determine the angle of rotation *φ* through the retroreflective targets in order to compare the results to the FDMS.

The data of the four distance meters of the FDMS were filtered in the first step, so only the measured distances on the respective planes were processed further. Here, 160,000 points were measured on each plane. Measurement values were filtered with a moving average as the large number of measured values was not necessary for the definition of a straight line/plane (Figure 12). The maximum standard deviation of the measured distances of a plane was 0.5 mm. Laser 4 hits a retroreflective target in almost all measurements; hence, the behaviour of the distance measurement is not reliable [41]. Therefore, the data of the fourth laser will not be considered. To rectify data to a plane, the direction vector, established with the PCO cameras, and the velocity of the torsion simulator were used. The difference between the first and last laser measurement point corresponds approximately to the height of the planes of the torsion simulator. Using the direction vector of the straight line and the direction vector of the rotation axis, the normal vector of each plane could be determined.

As additional control, 10 low-cost microelectromechanical systems sensors (MEMS sensors) were attached to the back of each plane (Figure 13). The achievable accuracy of the sensors was 0.1° [42]. MEMS data were averaged for each plane, resulting in tilt values around two coordinate axes describing the rotation angle *φ*. During data acquisition, noise was very high because the lift truck was instable, and the MEMS appeared to be too sensitive. Therefore, only the data before and after the movement were used to calculate a linear inclination change.

A tilt sensor Kern Nivel 20 was used as an inclinometer to provide reference data. The Nivel 20 is a high-precision instrument used to determine deviations from the horizontal plane [43]. The working range of the Nivel 20 was limited to ±0.11° with an accuracy of 0.02°. The entire object was rotated vertically, and then the individual planes were aligned horizontally with the Nivel 20 (Figure 14).

To define the plane using the FDMS data, the rotation axis $\vec{r}\;$ of the torsion simulator was required. A Nikon D850 camera was therefore used to take a photogrammetric set of images of the target in a static state. Using single measuring points describing the rotation axis, the direction vector could be calculated. Furthermore, the torsion simulator was measured by photogrammetry in the static state in order to determine the normal vector of the planes individually and to derive the rotation angles *φ*.

### 3.2. Comparison of the Rotation Angles φ

A total of 16 tests were performed, in which the torsion simulator was scanned with the laser beams in each case. The data sets were grouped into 5 groups. In group 1, all planes were vertical. In the other groups, planes 2, 3, and 4 were rotated. Plane 1 continued to serve as reference plane in the vertical position. For data group 2, the planes were rotated with the bottom edge toward the FDMS. For data groups 3–5, the bottom edge was rotated away from the FDMS. Planes were recorded repeatedly in the same state to determine the precision.

#### 3.2.1. Comparison between Data Sets

The absolute rotation angles could be calculated from the normal vectors, which could be determined with various measurement methods. Since all planes had the same rotation axis, no further parameters were necessary. For the absolute angle calculation, we used the normal of the very first data set (Equation (1)). Results are shown in Table 1. The calculated angle is displayed in green. The standard deviation s_RA_ of the repeated measurement is shown in yellow. Systematically, the calculated angles of the FDMS are smaller than those for MI. Deviations are smaller for the vertical planes than for the rotated planes. Also, the deviations are smaller for MI than for the FDMS. Differences between the data group with the vertical planes were minimal, with a maximum standard deviation of 0.02°, which proves the high precision of the distance meters. For the MI data, a standard deviation of 0.01° on average was calculated.

Subsequently, the angle discrepancy between the FDMS and MI was determined. The discrepancies sorted by the angle of rotation is shown in Figure 15. The difference was minimal with a value of 0.02° for the vertically oriented planes. For a rotation angle between 3 and 6 degrees, the deviations were 0.2°. The discrepancy increased to 0.3° with a rotation angle of 14°.

#### 3.2.2. Comparison within the Data Group

Another way to analyse the normal vectors of the planes was to calculate the angle difference within a data group. It represented the application on the wind turbine. Rotation angles between the planes for each data set were calculated. For the three planes considered here, three rotation angles could be determined (plane 1 to plane 2, plane 1 to plane 3, and plane 2 to plane 3). Results are shown in Table 2. Based on the repeated measurements of the different settings of the rotation angles within a data group, the standard deviation s_RA_ can also be calculated. Here, it can be seen that the standard deviations s_RA_ are small, and there is no difference between the FDMS and MI or between data groups with vertical and rotated planes.

In Figure 16, the discrepancy in the rotation angles between the FDMS and MI are shown. The discrepancies between plane 1 and plane 3 were constantly between 0.2° and 0.4°. If plane 2 was included, the discrepancies were larger and fluctuate. It was noticeable that, depending on the angular difference of the planes, a change of sign occurs in the deviations. For the interpretation, it was interesting to know that the planes for data group 2 were rotated in the other direction than for data groups 3, 4, and 5. Overall, the angle differences were higher in this comparison than in the comparison between data groups (Figure 15).

The image data of the torsion simulator in static state (Nikon D850 camera) also made it possible to derive rotation angles and to perform a comparison with the rotation angles of the MI and FDMS. A comparison with the MI data showed that the calculated angles differ by up to 0.3° for the rotated planes. For the vertical planes, the maximum difference was 0.08°. For the FDMS data, the difference was 0.3° for the vertical planes, but it was 0.6° for the rotated planes. This comparison confirms the results shown above.

Furthermore, it was possible to compare the data with the data of the MEMS sensors. Here, a comparison was made between the data sets of the individual sensors. The calculated inclinations after the adjustment with the Nivel 20 are 0°, confirming the good alignment of the planes with the Nivel 20. The standard deviation was 0.05° during the whole experiment. The calculated tilt within data group 1 (vertical plane) was between 0.00 and 0.04°, which confirms that no unwanted change of the rotation angles happened between the data sets. In a first check of the data sets with rotated planes, the MEMS data matched the MI data to within 0.01°. There were discrepancies of up to 0.3° in the FDMS data. These data confirm the previous results, so no further detailed analysis will be performed.

## 4. Discussion

Using the presented approach, it was possible to acquire the angle of rotation φ of a torsion simulator with the FDMS, which represented a possible torsion without contact or marking. To test the accuracy torsion that could be derived with the FDMS, the torsion simulator was developed. The focus was only on torsion. Other factors that occur on a rotor blade (for example, bending or the rounded shape) were not considered. In the tests, the torsion simulator was moved in a straight line, while a rotor blade rotates around a moving axis of rotation. Therefore, not all impact factors were taken into account, as they occur on a real wind turbine.

By means of a simulation, the calculation process could be checked, and the parameters influencing the determination of the torsion angle could be investigated. Here, it was shown that the relative orientation of the measuring system has a minor influence on the angle of rotation and that the influence of the distance measurement was decisive in determining the angle of rotation.

Acquiring real reference data was complex because a high level of accuracy was required. The tilt sensor met the criteria, but the planes could only be aligned vertically. Tilting the entire torsion simulator into different positions was possible to vary the angle of impact. After aligning the planes of the torsion simulator, repeat measurements were performed with the FDMS and MI. The calculated deviations were up to 0.02° with the MI (complete system) and up to 0.05° with the FDMS, confirming very high accuracy. According to the data sheet, it could be assumed that the achievable accuracy of the MI is higher than that of the FDMS. The FDMS is a multi-sensor system with a complex calibration and residual errors. Therefore, the deviation when comparing the four planes within one data set was higher than when comparing the data sets of individual planes. Further measurements with a larger variety of angular differences would help with interpretation. When evaluating the planes within a data set, the results for plane 2 were striking. Further investigations are necessary to determine whether it is due to the plane or the distance meter.

The calculated tosion angle depends in particular on the distance measurement. Therefore, regular calibration of the distance meters is recommended. The installation of more up-to-date distance meters is already being considered, and initial plans for the realisation are already underway.

## 5. Conclusions and Outlook

Non-contact, marker-free detection of rotor blade torsion during operation is a challenge. Especially given a desired precision of 1°, special methods had to be developed. The concept and first field tests that used a combination of photogrammetry and several laser scanners were presented in this work. A fan-shaped distance meter system (FDMS) was developed and tested. Using the developed torsion simulator, it was revealed that the FDMS detects rotation angles as small as 0.3°, with a standard deviation of about 0.05° for repeated measurements. On the assumption that the precision of the FDMS is constant also for larger distances, a torsion measurement with better than 1° resolution can be achieved.

However, this assumption should be tested in the field. In a first test, the quality of the data from the distance meters should be examined under real conditions. Furthermore, the nacelle movement should be recorded to make it possible to derive the torsion with the FDMS. This should be compared to wind turbines with built-in sensors in the rotor blade [44]. Alternatively, field tests on a turbine fitted with a random pattern [24] could lead to comparable data. These future investigations are addressed in an upcoming project, which is currently under review.

## Figures and Tables

**Figure 1 sensors-23-08603-f001:**
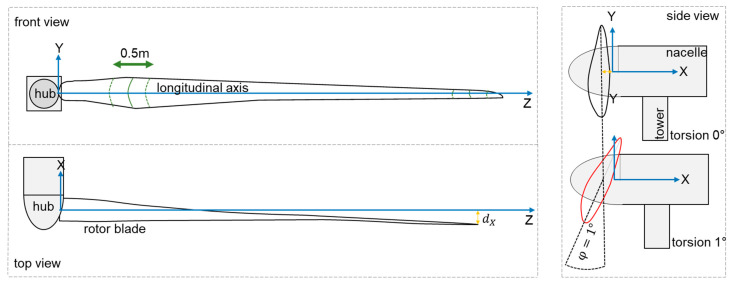
Top and front view of a rotor blade (**left**); estimation of the accuracy to be achieved for torsion determination of 1° (**right**).

**Figure 2 sensors-23-08603-f002:**
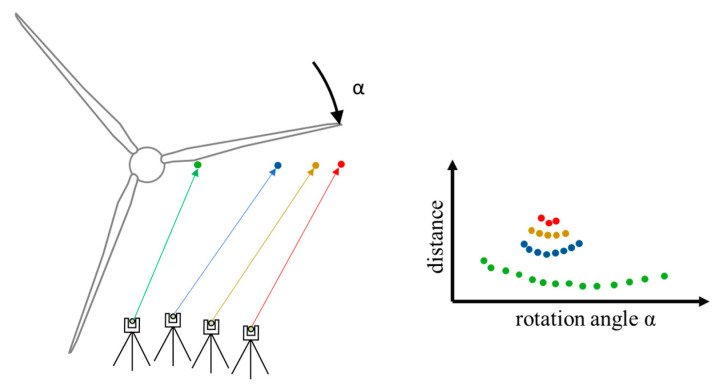
Recording of profile data with distance meters as a function of the rotation angle α.

**Figure 3 sensors-23-08603-f003:**
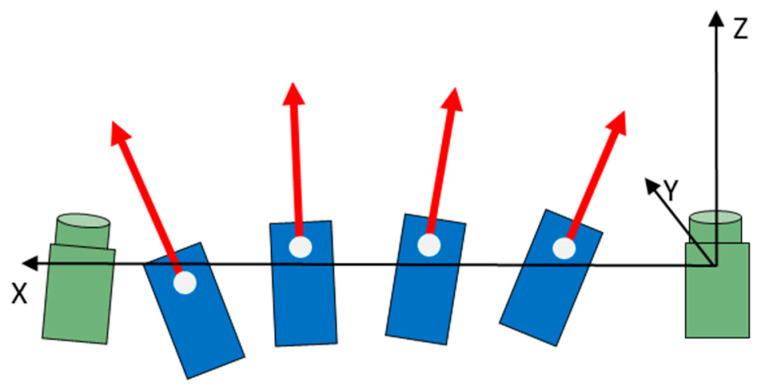
Concept of fan-shaped distance meter system in top view; laser beams (red arrows) form a plane.

**Figure 4 sensors-23-08603-f004:**
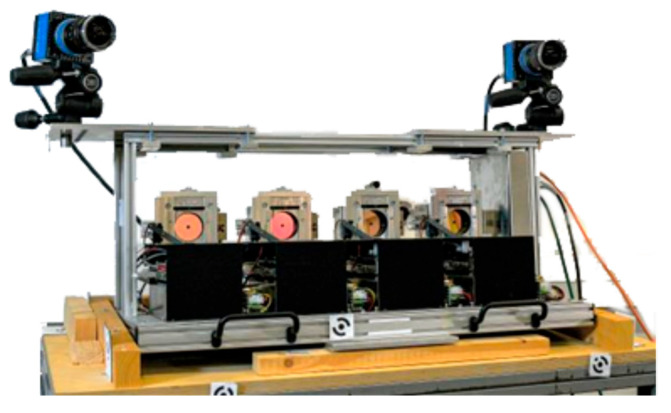
The FDMS with two cameras.

**Figure 5 sensors-23-08603-f005:**
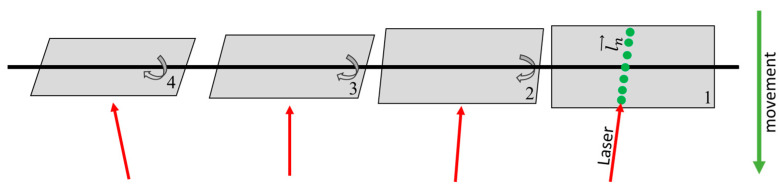
Torsion simulator with four planes that could be adjusted around a common axis of rotation. Planes 2–4 were rotated differently. Each laser measured on one plane.

**Figure 6 sensors-23-08603-f006:**
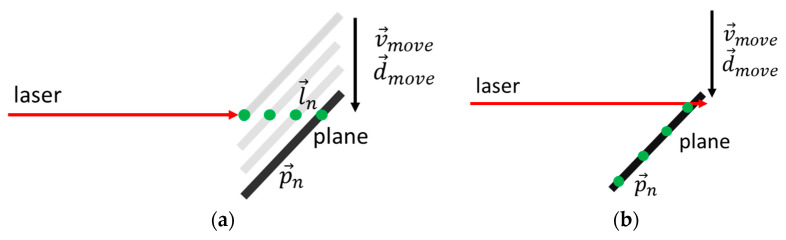
Procedures for the measurement and calculation of a laser on a plane: (**a**) By moving the torsion simulator, measured values (green) were acquired along laser beam; (**b**) measured values could be placed on the plane with a direction vector and velocity.

**Figure 7 sensors-23-08603-f007:**
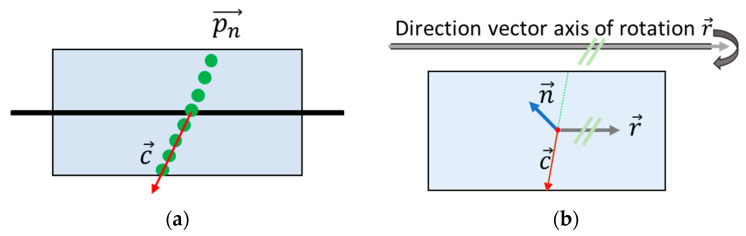
Calculation procedure: (**a**) Corrected measured 3D coordinates on plane in green and best-fit line $\vec{c}$ in red; (**b**) normal vector $\vec{n}$ of the planes could be determined by best-fit line $\vec{c}$ as well as the known axis of rotation of the plane $\vec{r}$.

**Figure 9 sensors-23-08603-f009:**
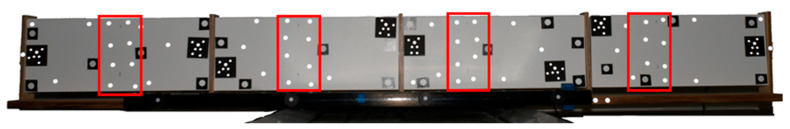
Torsion simulator with four planes for determination of the angle of rotation *φ*. The measuring path of the laser was located in the red rectangles. Retro-reflective points located within the red rectangles were used for calculation of the normal vector of the plane.

**Figure 10 sensors-23-08603-f010:**
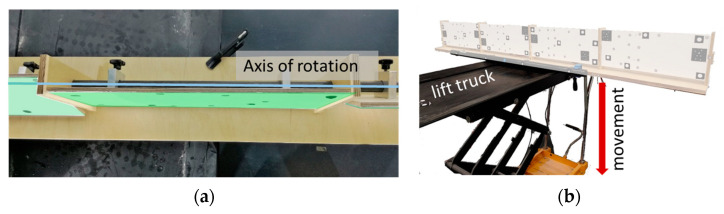
Torsion simulator: (**a**) Planes (green) could be rotated around a common axis of rotation (blue line); (**b**) for movement (red arrow), the torsion simulator stood on a lift truck.

**Figure 11 sensors-23-08603-f011:**
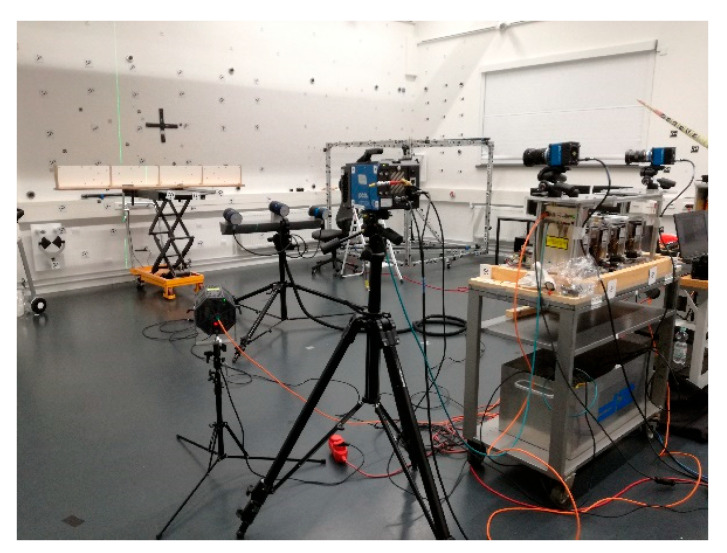
Setup of the measuring systems in the laboratory.

**Figure 12 sensors-23-08603-f012:**
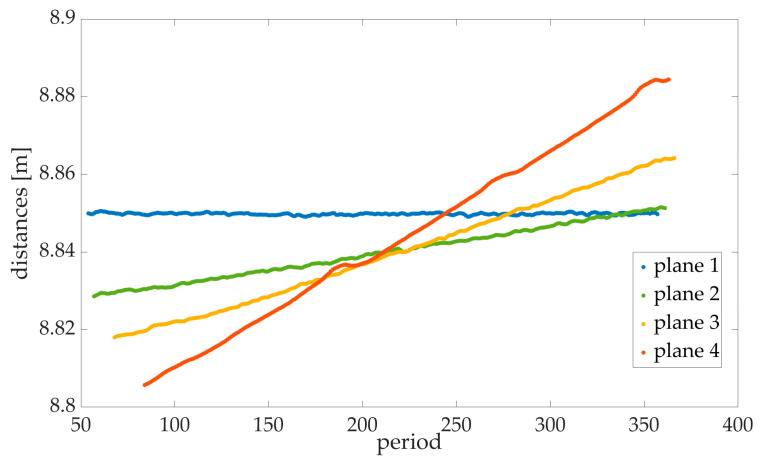
Filtered and averaged measured values of the FDMS for each frame on the planes. Plane 1 is vertical and, therefore, returns with the same distance for each measurement, while plane 4 is tilted most towards the horizontal. The period even allows to one see that, due to the movement from top to bottom, the bottom side of the plane is towards the top side away from the FDMS.

**Figure 13 sensors-23-08603-f013:**
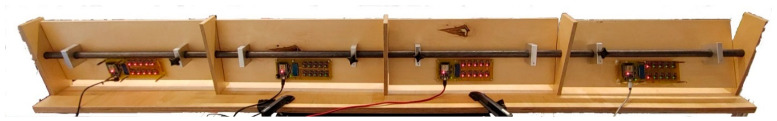
MEMS attached to the back of the planes.

**Figure 14 sensors-23-08603-f014:**
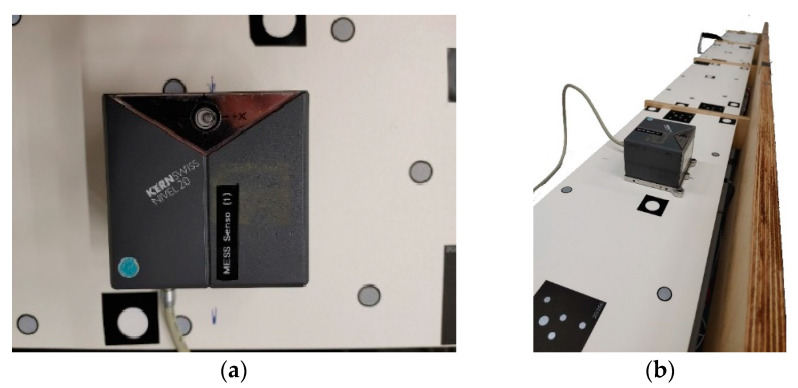
Alignment with the tilt sensor Nivel 20; (**a**) tilt sensor in the working range; (**b**) planes in the horizontal position.

**Figure 15 sensors-23-08603-f015:**
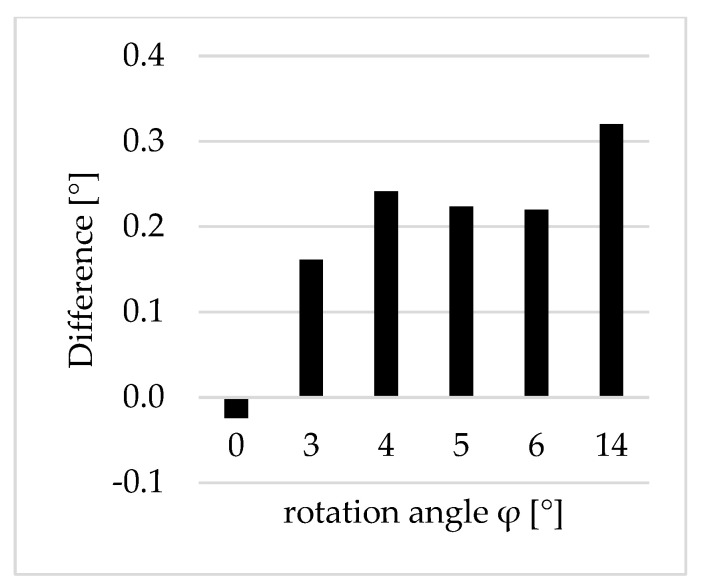
Differences in the angles [°] between the data sets of the FDMS and AICON MoveInspect.

**Figure 16 sensors-23-08603-f016:**
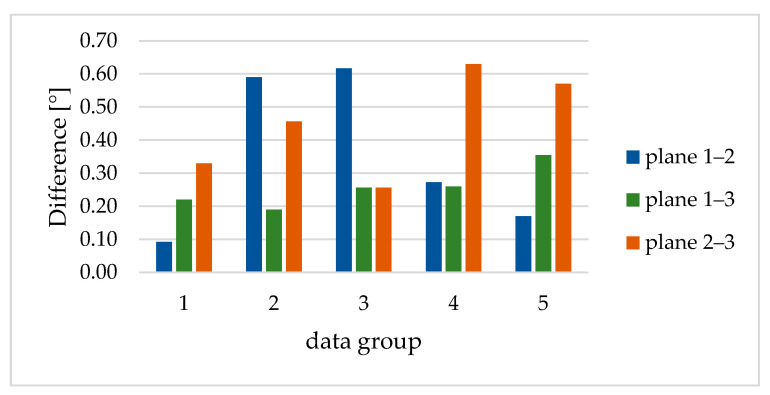
Differences in the angles [°] between the planes of respective data groups.

**Table 1 sensors-23-08603-t001:** Results of measurements obtained with the torsion simulator. For each plane, the angle between the normal vectors was calculated and displayed in green. The reference in each case was data set 1. Furthermore, the standard deviation (yellow) within the data groups with the same alignment is reported.

Plane		Data Group				
		Vertical	Rotated			
		1	2	3	4	5
**1**	Average FDMS [°]	0.02	0.01	0.11	0.06	0.07
	Average MI [°]	0.01	0.03	0.05	0.03	0.08
	S_RA_ FDMS [°]	0.00	0.03	0.02	0.02	0.03
	S_RA_ MI [°]	0.00	0.00	0.00	0.00	0.01
**2**	Average FDMS [°]	0.05	3.05	3.11	1.59	6.35
	Average MI [°]	0.02	3.21	3.24	1.69	6.57
	S_RA_ FDMS [°]	0.02	0.02	0.03	0.03	0.06
	S_RA_ MI [°]	0.01	0.01	0.01	0.01	0.00
**3**	Average FDMS [°]	0.03	3.08	5.48	4.82	13.59
	Average MI [°]	0.01	3.28	5.70	5.06	13.91
	S_RA_ FDMS [°]	0.01	0.03	0.06	0.06	0.11
	S_RA_ MI [°]	0.00	0.05	0.01	0.01	0.01

**Table 2 sensors-23-08603-t002:** Results of measurements with the torsion simulator within a data group. The average of the angle difference between two planes is shown in green. The standard deviations (yellow) of the data groups with the same alignment are shown.

Plane		Data Group				
		Vertical	Rotated			
		1	2	3	4	5
**1–2**	Average FDMS [°]	0.49	2.52	2.50	2.13	6.92
	Average MI [°]	0.40	3.11	3.12	1.86	6.75
	S_RA_ FDMS [°]	0.04	0.00	0.05	0.03	0.09
	S_RA_ MI [°]	0.01	0.01	0.00	0.01	0.00
**1–3**	Average FDMS [°]	0.35	2.69	5.02	5.22	14.01
	Average MI [°]	0.57	2.88	5.28	5.48	14.37
	S_RA_ FDMS [°]	0.02	0.06	0.08	0.06	0.14
	S_RA_ MI [°]	0.09	0.05	0.02	0.00	0.00
**2–3**	Average FDMS [°]	0.14	0.18	2.52	3.09	7.09
	Average MI [°]	0.47	0.63	2.27	3.72	7.66
	S_RA_ FDMS [°]	0.02	0.05	0.03	0.03	0.05
	S_RA_ MI [°]	0.08	0.02	0.02	0.01	0.01

## Data Availability

Not applicable.

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
