# Peer review of "Development of a Procedure for Torsion Measurement Using a Fan-Shaped Distance Meter System"

_sensors, 2023, doi:10.3390/s23208603_

Round 1

Reviewer 1 Report

      This paper research the marker-free and contactless measurement of rotor blades during operation, which is useful for maximising the efficiency of wind turbine. An interesting and nice research work! 

       A minor comment: please conduct some editing to avoid the grammar error and make the language more authentic.

A minor comment: please conduct some editing to avoid the grammar error and make the language more authentic.

Author Response

Thank you very much for taking the time to review this manuscript. We have checked the text and made some small changes to the grammar.

Reviewer 2 Report

I have read this article with great interest. The manuscript is well-structured, and logically sound, but there are also some shortcomings. I have provided some minor revision suggestions, which can be found in the attached PDF.

Author Response

Thank you very much for taking the time to review this manuscript.

Response 1:

Thank you for pointing this out. We agree with this comment. The description has been expanded to better understand how to determine the 8mm measurement accuracy.

“At the outer tip, the blade has a blade depth of 1m. If a torsion φ of 1°around the centre of the blade is assumed, distance measurement must achieve an accuracy of 8mm (blade depth of 0.5m) via arc formula (Figure 1, side view). These requirements can be met with laser scanners or distance meters.”

Response 2:

The measurement on level 4 unfortunately caused problems with the retroreflective target. However, the measurement setup and evaluation are very complex and time consuming, so the measurements cannot be repeated for this paper. However, it is intended that these measurements will be repeated in the future and hopefully confirm the measurement accuracy achieved. This is the subject of the discussion section. Further measurement objects will follow as well.

I would like to present the data of laser 4 in figure 12 to show the effects of the retroreflecting targets on the distance.

Response 3:

In the article, the angles are only considered absolute, because it is only about the accuracy of the angle. However, the direction of the angle can also be determined from the measured distances. From the figure 12, the conclusions on the direction can already be drawn.

Response 4:

In the article many publications are presented to past procedures for the measurement of wind turbines. The topic is partly very special, so that little current literature is available. If you know of current literature, please feel free to contact me.

Reviewer 3 Report

Review for paper sensors-2618930 named

Development of a Procedure for Torsion Measurement Using a Fan-Shaped Distance Meter System

In this paper, authors show an approach for  marker-free and contactless measurement of rotor blades torsion deformations during the motion.

An innovative measurement system called the Fan Distance Measuring System (FDMS) was applied to the deformation measurement, which uses a combination of deformation measurement using multiple laser scanners and photogrammetry.

I suggest making small corrections to improve the quality of the work:

In Figure 2 what are the target points? The green dot is looking for the green zone from Figure 1 and the red one aims at the end of rotor blade. Please explain picture 2 in more detail.

What does the laser layout in Figure 3 show? How did you come up with that schedule?

Line 202 authors said „A best-fit line ?⃗ could be determined by calculated coordinates ?⃗?“. Explaine what is ?⃗?.

Equation 2: ?⃗⃗?°?⃗⃗? here circle is scalar product of two normal vectors. Remove the circle or place a dot.

Figure 10: Here it is intended to observe the translational movement of rotor blades. It is clear that in laboratory conditions you cannot achieve realistic rotation from a rotor blade. In reality, the rotor blade rotates around the shaft. How do you justify this simplification of movement? Is there any relation between translation and rotation?

Figure 12: period is seconds or frames?

Figure 13: In a real situation, how would you attach MEMS  to the back of the planes/rotor blades?

The paper presents a new and interesting approach for the measurement of torsion of the rotor blade. believe that corrections should be made that would clarify the experiment to all readers of the paper, and I suggest to the editors to accept it for publication after the corrections.

Author Response

Thank you very much for taking the time to review this manuscript.

Response 1:

Figure 2 is now described in more detail in the article, so it should be understandable how data is recorded.

“All distance meters are aligned to the height of the hub and distributed along the blade (Figure 2). The green point measures directly at the hub and the red one at the outer tip. Further points (here in blue and yellow) can be positioned in between. Due to rotation of the rotor blade (rotation angle α), different distances are detected by the distance meters, which depend on rotation angle α. This enables an angle-based assignment of the measured values. In 1D mode, distances are detected according to the rotation angle of the rotor blades or the time in measurement system. Data recording at the hub takes longer than at the outer tip because the blade shape and rotation speed are different. Intensity values of detected laser beam are used to filter the data.”

Response 2:
Figure 3 on the concept of FDMS is now described in a bit more detail. However, for details on the development, I would like to kindly refer to the article mentioned there.

“In order to optimize the laser scanner approach, a novel measurement system has been developed, called a fan-shaped distance meter system (FDMS). The concept is shown in Figure 3. Four distance meters (blue rectangle) are used for this purpose, whose laser beams form a plane. This system aims to simplify the alignment of the la-ser beam to rotor blades, improves precision of the orientation, and provides a cost-effective alternative to laser scanners. Details about the FDMS were presented in [37].”

Response 3:
In line 202, the vector p is now also explained in the text, which should help for understanding.

“Since measured distances lay on laser line l_n, they had to be rectified to correctly reproduce the torsion simulator (Figure 6). Distances could be corrected using a known velocity v ⃗_move  and the direction vector d ⃗_move of the movement. Result are the points p ⃗_n on the plane. Thus, it was possible to convert 3D coordinates into a metric and time-independent system.”

Response 4:

Formula 2 shows the general formula for calculating two angles between vectors. I have checked the formula and found no error. The calculation of the scalar product is correct.

Response 5:
The torsion simulator of figure 10 should be the first step for simplifying the measurement problem. Herewith it is possible to conclude the achievable accuracy of the FDMS. For further laboratory tests we have a model of a wind turbine at our disposal, so that the rotation around the hub can be done. The speed is different at each measuring point. In the laboratory, the velocities and motions can be calculated by photogrammetric methods with high accuracy.

Response 6:
In Figure 12, 'period' describes the frames. A note has been added to the text.

Response 7:
I would not install MEMS on a real wind turbine to determine torsion. In essence, the MEMS are likely to be installed similarly to strain gauges. I would determine comparative data there photogrammetrically (by attaching measurement markers or a random pattern) as described in the conclusion and outlook.